


Published on behalf of



# Using bioluminescence as a tool for studying diversity in marine zooplankton and dinoflagellates: an initial assessment

Francis Letendre, Abigail Blackburn, Ed Malkiel and Michael Twardowski

Harbor Branch Oceanographic Institute, Florida Atlantic University, Fort Pierce, FL, United States of America

## ABSTRACT

Bioluminescence is light chemically produced by an organism. It is widespread across all major marine phyla and has evolved multiple times, resulting in a high diversity of spectral properties and first flash kinetic parameters (FFKP). The bioluminescence of a system is often a good proxy for planktonic biomass. The species-specific parameters of bioluminescent displays can be measured to identify species *in situ* and describe planktonic biodiversity. Most bioluminescent organisms will flash when mechanically stimulated *i.e.*, when subjected to supra-threshold levels of shear stress. Here we compare first flash kinetic parameters such as flash duration, peak intensity, rise time, decay time, first-flash mechanically stimulated light and e-folding time obtained with the commercially available Underwater Bioluminescence Assessment Tool (UBAT). We provide descriptions of the first flash kinetic parameters of several species of dinoflagellates *Pyrocystis fusiformis*, *Pyrocystis noctiluca*, *Pyrodinium bahamense*, *Lingulodinium polyedra*, *Alexandrium monilatum* and two zooplankton (the ctenophore *Mnemiopsis leidyi* and the larvacean *Oikopleura sp.*). FFKPs are then compared and discussed using non-parametric analyses of variance (ANOVAs), hierarchical clustering and a linear discriminant analysis to assess the ability to use bioluminescence signatures for identification. Once the first flash kinetic parameters of a bioluminescent species have been described, it is possible to detect its presence using emissions collected by *in situ* bathyphotometers. Assessing abundance and diversity of bioluminescent species may therefore be possible.

## INTRODUCTION

Bioluminescence is the emission of light through luciferin-luciferase mediated oxydation by a living organism. It is widespread across several clades, from dinoflagellates to copepods, gelatinous plankton and bony fishes (*Haddock, Moline & Case, 2010*; *Herring, 1983*; *Herring, 1987*). *Martini & Haddock (2017)* found that nine out of the 13 taxonomic categories studied were mainly bioluminescent, *e.g.*, 97% of cnidarians observed, whereas 100% of the ctenophores observed by *Morin (1983)* were found to be bioluminescent. Several species of calanoid copepods are bioluminescent and are thought to be the source of coelenterazine, a form of luciferin, for their predators *e.g.*, cephalopods, fish and cnidarians (*Takenaka, Yamaguchi & Shigeri, 2017*). In fact, presence of bioluminescence

Corresponding author
Francis Letendre, fletendre@fau.edu

in such diverse taxonomic groups suggests the trait evolved independently at least 40 times (*Haddock, Moline & Case, 2010*). Given the diversity of ecological uses and wide range in organism types with varying levels of evolved sophistication, it may be reasonable to postulate a diversity in bioluminescence expressions that may serve as a convenient diagnostic for identification.

Bioluminescence may serve several defensive, predatory and reproductive functions. These include but are not limited to camouflage *via* counterillumination (*Jones & Nishiguchi, 2004*), intraspecific warning signals (*Buskey & Swift, 1985*; *Takenaka, Yamaguchi & Shigeri, 2017*), the attraction of a predator-of-a-predator to protect oneself from a specific threat (*Abrahams & Townsend, 1993*) and aposematism (*Jones & Mallefet, 2013*). A key function of bioluminescence in dinoflagellates can be explained by the burglar alarm hypothesis (*Abrahams & Townsend, 1993*). When a dinoflagellate cell is entrained in a feeding current of a zooplankton, the shear stress associated with this current mechanically stimulates the organism. This bioluminescent emission then attracts larger predators which can prey on the zooplankton originally trying to feed on dinoflagellates. *Davis et al. (2014)* showed displaying bioluminescence in deep sea teleost fish might increase speciation events.

While it is clear bioluminescence serves many purposes for the individual producing the light, it can also provide information for scientists studying marine ecosystems. Indeed, measuring the bioluminescence of key species can provide us with insight on marine ecosystem health and its physical properties, *e.g.*, rise of harmful algal blooms populations, population dynamics and productivity, biodiversity assessments and fish school composition and size (*Altınağaç et al., 2010*; *Johnsen et al., 2014*; *Kim et al., 2006*; *Messié et al., 2019*). *Lieberman et al. (1987)* found bioluminescence intensity to be inversely correlated to surface water temperature for large scale areas, and positively correlated to chlorophyll at smaller scales, providing information on the productivity of an ecosystem. *Neilson, Latz & Case (1995)* also found a similar link between bioluminescence and chlorophyll measurements in the North Atlantic, only they found it to be highly seasonal with a high correlation in late spring. *Craig et al. (2010)* found a strong positive correlation between surface chlorophyll $\alpha$ and bioluminescence density in the 500–1,000 m depth range. On the other hand, *Buskey (1992)* found a correlation between bioluminescence and zooplankton biomass in the Greenland Sea, but no link could be established with any other environmental variables. It is also possible to identify thin layers and copepod aggregation through measuring bioluminescence activity through a water column (*Haddock, Moline & Case, 2010*). Indeed, *Widder et al. (1999)* associated peak bioluminescent emissions with high concentrations of the bioluminescent copepod *Metridia lucens* located in very thin layers in the Gulf of Maine.

Bioluminescence has been used to discriminate between dinoflagellates and zooplankton groups and abundances in the water column by comparing their flash kinetics (*Moline et al., 2009*) and overall bioluminescent intensity (*Swift et al., 1983*). Research on bioluminescence has provided a base for detecting and monitoring ctenophore populations (*Widder et al., 1999*), harmful algal blooms (*Haddock, Moline & Case, 2010*) and global ocean health (*Piontkovski & Serikova, 2022*).
Bioluminescent plankton emit light mainly through mechanical stimulation (*Haddock, Moline & Case, 2010*; *Letendre et al., 2024*). Mechanically stimulated bioluminescence (MSL) is produced when a supra-threshold level of shear stress is applied to the membrane of the cell, which is species-specific (*Latz, Nauen & Rohr, 2004*). Past the threshold, peak intensities of bioluminescence emissions are positively correlated to shear stress levels until maximal emission is reached (*Christianson & Sweeney, 1972*; *Deane & Stokes, 2005*; *Rohr, Losee & Hoyt, 1990*). However, the shear stress thresholds of bioluminescent organisms are usually unknown, with most studies to date focusing on dinoflagellates (*Cussatlegras & Le Gal, 2007*; *Latz, Nauen & Rohr, 2004*; *Lutz, Case & Gran, 1994*; *Maldonado & Latz, 2007*; *Rohr et al., 1997*).

At the mesoscale, prevailing bioluminescent organism distributions are influenced by many factors, *e.g.*, nutrient availability, seasonality, upwelling dynamics, water pollution. Latitude and seasonality can influence MSL of the water column (*Cronin et al., 2016*). The diel vertical migration and spatial heterogeneity of zooplankton also significantly affect water column MSL (*Tokarev et al., 1999*).

Intra-specific variation of MSL also exists and environmental pressure can affect bioluminescence of individuals. Indeed, in the presence of chemical cues from copepods, the MSL of the dinoflagellates *L. polyedra* and *A. tamarense* significantly increases (*Lindström et al., 2017*). For the invasive ctenophore *Mnemiopsis leidyi*, variables such as life stage (*Tokarev, Mashukova & Sibirtsova, 2012*), organism size and water temperature (*Tokarev & Mashukova, 2016*), concentration of heavy metals in the water (*Mashukova, Tokarev & Skuratovskaya, 2017*), injuries (*Mashukova & Tokarev, 2016*), and diet (*Mashukova & Tokarev, 2013*) have all been found to have an impact on bioluminescence emissions.

When mechanically stimulated, first flash emissions follow similar kinetic patterns (*Latz & Rohr, 1999*; *Widder & Case, 1981*). Bioluminescent emissions can be described quantitatively using parameters such as rise time (RT), decay time (DT), flash duration (FD), peak intensity (PI), first-flash mechanically stimulated light(FF-MSL) and e-folding time (EF) (Fig. 1). Rise time (ms) is measured as the time between the first signal above the instrument's baseline to the highest instantaneous photon emission *i.e.,* the peak intensity (photons s$^{-1}$). Decay time (ms) is measured as the emission time between the peak intensity and the return to baseline following an exponential decay. The flash duration (ms) is the total time of the bioluminescent signal above the baseline. The FF-MSL (photons flash$^{-1}$) is the total amount of photons emitted during that flash duration. e-folding time (ms) is defined as the decay time from the peak to 1/e of the peak intensity.

Since the first flash kinetic parameters (FFKPs) can vary greatly among bioluminescent planktonic species (*Letendre et al., 2024*), light emissions can facilitate plankton identification, in some cases autonomously (*Cronin et al., 2016*; *Johnsen et al., 2014*; *Moline et al., 2009*; *Nealson, Arneson & Huber, 1986*). Simultaneous use of bathyphotometers and fluorometers can further discriminate between autotrophic and heterotrophic bioluminescent plankton (*Messié et al., 2019*).

Herein, we describe the first-flash kinetics of *Pyrodinium bahamense* and *Alexandrium monilatum*, two dinoflagellate HAB species of the Indian River Lagoon (IRL), Florida,

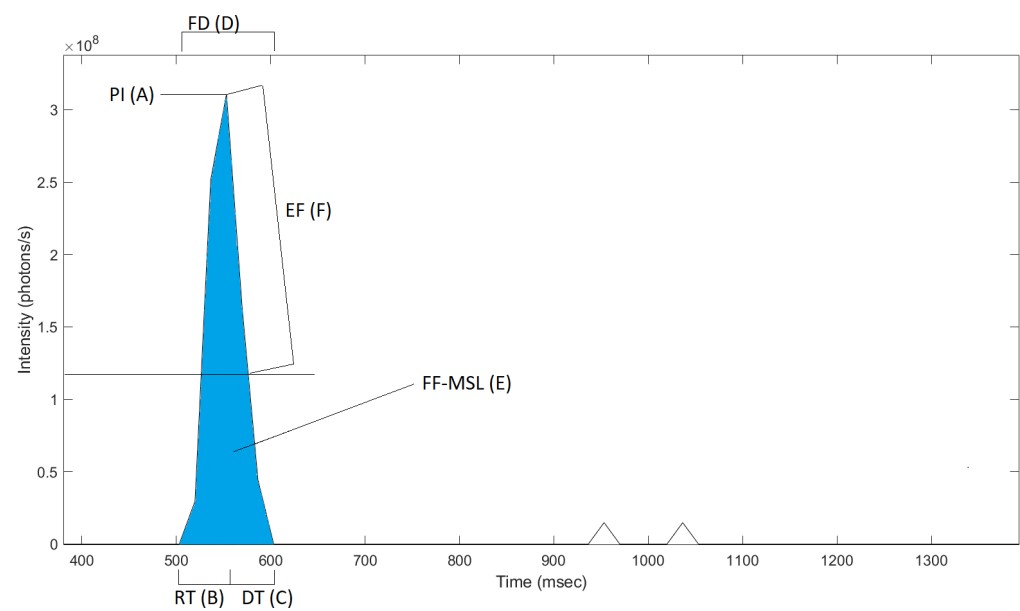

**Figure 1** **Typical first flash kinetic parameters of the dinoflagellate *Pyrodinium bahamense* obtained through shear-induced mechanical stimulation in the UBAT bathyphotometer.** (PI) Peak intensity, (RT) Rise time, (DT) Decay time, (FD) Flash duration, (FF-MSL) First-flash mechanically stimulated light and (EF) e-folding time. *Letendre et al. (2024)* shows a typical first flash response of the dinoflagellate *Pyrocystis fusiformis*.

along with the dinoflagellates *Pyrocystis fusiformis*, *Pyrocystis noctiluca* and *Lingulodinium polyedra*. The ctenophore *Mnemiopsis leidyi* and the larvacean *Oikopleura sp.* were additionally two zooplankton tested that are often found in the IRL. These species were selected to provide a wide range of bioluminescent emissions, both in intensity and time parameters. FFKPs were obtained using a UBAT, a commercially available bathyphotometer (http://www.seabird.com; Seabird Scientific, Bellevue, WA, USA), a mixing chamber where FFKPs may be reliably resolved but subsequent flashes are ambiguous based on ranges of shear experienced in the chamber and variable residence times (*Thombs, Shulman & Matt, 2024*). Statistical analyses were made to pinpoint which FFKPs can be used to identify bioluminescent species based on their emission characteristics. A linear discriminant model was developed in order to test the ability to identify bioluminescent species solely based on their emissions. This study provides insight on how bioluminescence can be used to monitor and assess marine biodiversity, while providing fundamental and novel data on dinoflagellate and zooplankton FFKPs.

## METHODS

### Culture maintenance and species provenance

Cultures of *Lingulodinium polyedra* (CCMP1738) and *Pyrocystis noctiluca* (CCMP732) were obtained from The National Center for Marine Algae and Microbiota (NCMA) at Bigelow Laboratory. A culture of *Pyrocystis fusiformis* was purchased from PyroFarms, California. Individuals of *Pyrodinium bahamense* and *Alexandrium monilatum* were isolated from a

phytoplankton net tow (20 mesh) made in the Indian River Lagoon, Florida. All species were then cultured in L1/2-Si medium in a growth room at a constant temperature of 23 °C. Inoculates were transferred every two weeks during their log-growth phase for semi-continuous culturing. Cultures were put under a reverse day/night illumination cycle of 12h:12 h at 80 μmol photons m$^{-2}$s$^{-1}$ to insure the scotophase was during the day and allow for daytime bioluminescence experiments. All cultures were not axenic.

Adult ctenophores *Mnemiopsis leidyi* and larvaceans *Oikopleura sp.* were collected at incoming tides in the Harbor Branch Oceanographic Institute channel using a 153 μm zooplankton net, or using a pierced cup mounted on a stick. Single ctenophores were then transferred into individual beakers filled with 0.22 μm filtered seawater from the HBOI channel. Beakers were immediately put into a dark room for dark adaptation before testing in UBAT. Individual larvaceans were placed into caps of centrifuge tubes filled with FSW and immediately dark-adapted. The dark adaptation period for zooplankton species was four hours.

## Instrumentation, radiometric calibration and bioluminescence measurements

Bioluminescence emissions were measured using the Underwater Bioluminescence Assessment Tool (http://www.seabird.com; Seabird Scientific, Bellevue, WA, USA). The UBAT's internal flow of 0.3 L s$^{-1}$ mechanically stimulated organisms through a range of shear stress levels using an impeller directly upstream of the 0.44 L mixing chamber (*Orrico et al., 2009*). Organisms are stimulated in turbulent flow in the integrating chamber for up to 10 s, where a PMT light detector samples at 60 Hz.

For accurate measurements of first flash kinetic parameters of individuals dinoflagellates, cells were isolated following several dilutions in filtered seawater. Then, single organisms were picked using a cell sucker and gently released into caps of five mL centrifuge tubes. For the chain forming *A. monilatum*, only individual cells were isolated and selected. Tested organisms were selected from the newest transfers/generations, increasing probability of testing individuals of similar sizes and ages within one species. Following cell isolation, centrifuges caps were put into a dark room 2 h prior testing for adequate dark adaptation. The room was made lightproof by blocking any stray light from the door using black felt. The UBAT was submerged into a black bin filled with filtered seawater at 23 °C and salinity matching the culture medium. A black lid was placed on the bin to block stray light while leaving enough room to introduce organisms. A 10 μm nitex mesh was attached to the output of the UBAT, preventing recirculation of cells. Single cells were introduced gently into the UBAT's intake flow one by one and only after a positive signal *i.e.,* a bioluminescent flash, was recorded to prevent multiple simultaneous emissions. First flash kinetic parameters, *e.g.,* rise time, decay time, e-folding time, flash duration, peak intensity and FF-MSL (Fig. 1), were analyzed in MATLAB. Noise filtering was made by fixing a $1.5 \times 10^7$ photons/s signal threshold. Peak intensities were measured using the findpeaks MATLAB function. When the 60 Hz sampling rate of the UBAT did not fully capture the start or the end of the flash, interpolation were made to estimate timing FFKPs. When testing adult ctenophores, obvious pre-stimulation was observed at the UBAT's

intake during the organism's entrance into the light baffle. It is unclear how much light is lost from the first flash emission before the ctenophore reaches the integrating cavity. This type of pre-stimulation was not observed when testing dinoflagellates.

The UBAT was calibrated before every experiment using a LED calibration device with constant emission provided by the manufacturer. However, independent calibration with a cylindrical four mm x two mm phosphorescent tritium emitter (450 nm with a 150 full width at half maxima), previously calibrated radiometrically in our lab with an integrating sphere apparatus coupled with a power meter, determined the calibration scaling factor with the LED device was a significant 2.5 times too low (*Blackburn et al., 2023*). This correction factor was applied to all MSL measurements from the UBAT. Implications for this finding are assessed in the Discussion section.

### Statistical analyses

Shapiro–Wilk tests were made for all first flash kinetic distributions of dinoflagellates and zooplankton species tested. All distributions did not satisfy the normality condition. Considering non-normality, non-parametric bootstrapping with 500 iterations was computed for all FFKPs to generate 95% confidence intervals. A non-parametric ANOVA was done for each kinetic, *i.e.,* PI, RT, DT, FD, FF-MSL and EF, using the Kruskall-Wallis test. This allowed assessment of the FFKPs that may be used to effectively differentiate between species from their bioluminescence emissions.

To visualize similarities in bioluminescence emissions among species and compare with the actual phylogeny, a hierarchical clustering analysis using FFKP averages was made and then compared to the phylogeny of tested species (*Kassambara, 2017*). Following a factor analysis to identify key variables and optimize model accuracy, a linear discriminant analysis was made using all tested species except *P. bahamense*, and all flash kinetics except FF-MSL and DT. All analyses were done in RStudio 4.2.2 (*Tharwat et al., 2017*).

## RESULTS

The distribution of rise time, decay time, e-folding time and flash duration for all tested species are compiled in boxplots in Fig. 2. *M. leidyi* was consistently the species with the highest and most variable parameters. Rise times for dinoflagellate species were generally under 100 ms, with the exception of a few *Pyrocystis fusiformis* samples reaching over 150 ms. For dinoflagellates, decay times were usually under 1 s, however the distribution of *P. fusiformis* in some cases was >1 s. For dinoflagellate species, e-folding time was the time parameter with the highest intra-specific variability (Fig. 2C). The ctenophore *M. leidyi* had the widest distributions across all time FFKPs, which was possibly caused by higher variability in its size than the single cell dinoflagellates and by the presence of multiple simultaneous sources of stimulation (see Discussion). First flashes of *M. leidyi* rarely reached more than 2 s in length (Fig. 2D). All species of dinoflagellates had emissions under 1 s. FFKPs of the larvacean *Oikopleura sp.* had statistically similar distributions to the dinoflagellates *Pyrodinium bahamense* and *A. monilatum*. However, its e-folding had a wider distribution than all dinoflagellates. Moreover, *P. bahamense*, *L. polyedra* and *A. monilatum* generally had very similar distributions for time-dependent FFKPs. Peak

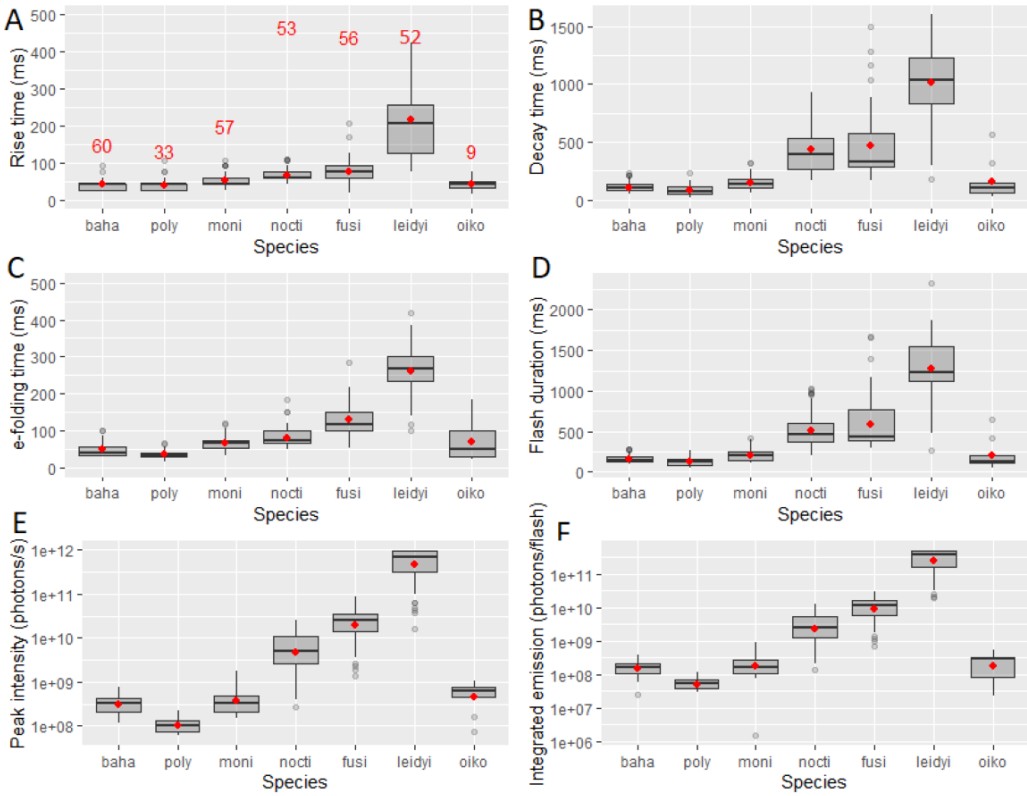

**Figure 2** Boxplots showing the first flash (A) Rise time, (B) Decay time, (C) e-folding time, (D) Flash duration (E) Peak intensity and (F) Integrated emission of the five species of dinoflagellates, the ctenophore *Mnemiopsis leidyi* and the larvacean *Oikopleura* sp. that were mechanically stimulating within the UBAT. The number of observations are displayed on frame (A) and are consistent throughout all frames. Respective means are displayed on the distributions as a red circle. Measurements two standard deviations above or below the mean are displayed as outliers. Species abbreviations are as follows *Pyrodinium bahamense* (baha), *Lingulodinium polyedra* (poly), *Alexandrium monilatum* (moni), *Pyrocystis noctiluca* (nocti), *Pyrocystis fusiformis* (fusi), *Mnemiopsis leidyi* (leidyi) and *Oikopleura sp.* (oiko).

intensities averages ranged from $1 \times 10^8$ to $1 \times 10^{12}$ photons/s and FF-MSL ranged from $1 \times 10^7$ to $1 \times 10^{11}$ photons/flash, spanning six orders of magnitude across case species for both FFKPs (Figs. 2E–2F).

Table 1 summarizes all FFKPs that were measured with the UBAT. Bootstrapped confidence interval (95%) were computed in RStudio 4.2.2 from the tested samples. All species had at least 30 individuals tested, with the exception of *Oikopleura sp.*, due to its very low abundance at the collection site and its low survival and flash rate. Peak intensities spanned 4 orders of magnitude, with the dimmest species being *Lingulodinium polyedra* at $2.68 \times 10^8 \pm 1.08 \times 10^8$ photons/s and the brightest being *M. leidyi* at $1.60 \times 10^{12} \pm 8.71 \times 10^{11}$ photons/s on average. Since FF-MSL is directly influenced by peak intensity, as it is the total amount of photons emitted in the first flash, similar trends were observed. *L. polyedra* emitted the least amount of photons per flash at $7.98 \times 10^7 \pm 5.21 \times 10^7$ photons[1] flash[−1] and the ctenophore emitted the most at $8.10 \times 10^{11} \pm 4.31 \times 10^{11}$ photons[1] flash[−1] on average.

**Table 1 First flash kinetic parameters of all tested species acquired through mechanical stimulation in the UBAT.** The upper values indicate bootstrapped 95% confidence interval and the lower values indicate the distribution's mean and standard deviation. Sample sizes are indicated next to the species' name.

| Species | Flash kinetics | | | | | |
|---|---|---|---|---|---|---|
| | Rise time (ms) | Decay time (ms) | Flash duration (ms) | e-folding time (ms) | Peak intensity (photons s$^{-1}$) | FF-MSL (photons$^{1}$flash$^{-1}$) |
| *Pyrocystis fusiformis (56)* | 69.1–84.3 | 424–568 | 529–605 | 117–142 | $5.51 \times 10^{10}$–$7.82 \times 10^{10}$ | $2.73 \times 10^{10}$–$3.37 \times 10^{10}$ |
| | $76.8 \pm 31.0$ | $513 \pm 328$ | $536 \pm 288$ | $129 \pm 47.5$ | $6.64 \times 10^{10} \pm 4.48 \times 10^{10}$ | $2.91 \times 10^{10} \pm 1.79 \times 10^{10}$ |
| *Pyrocystis noctiluca (53)* | 59.9–69.6 | 382–494 | 455–573 | 74.6–88.9 | $1.37 \times 10^{10}$–$2.18 \times 10^{10}$ | $6.78 \times 10^{9}$–$1.07 \times 10^{10}$ |
| | $65.0 \pm 18.4$ | $442 \pm 211$ | $515 \pm 222$ | $81.7 \pm 27.2$ | $1.80 \times 10^{10} \pm 3.7 \times 10^{8}$ | $8.82 \times 10^{9} \pm 7.26 \times 10^{9}$ |
| *Pyrodinium bahamense (60)* | 38.4–45.2 | 98.1–120 | 148–172 | 44.8–54.1 | $7.50 \times 10^{8}$–$9.37 \times 10^{8}$ | $3.59 \times 10^{8}$–$4.56 \times 10^{8}$ |
| | $42.0 \pm 13.9$ | $109 \pm 44.3$ | $160 \pm 49.3$ | $49.9 \pm 18.5$ | $8.44 \times 10^{8} \pm 3.71 \times 10^{8}$ | $4.41 \times 10^{8} \pm 6.06 \times 10^{7}$ |
| *Lingulodinium polyedra (33)* | 32.6–45.2 | 69.9–102 | 114–151 | 33.0–42.1 | $2.31 \times 10^{8}$–$3.05 \times 10^{8}$ | $1.15 \times 10^{8}$–$1.51 \times 10^{8}$ |
| | $39.7 \pm 18.9$ | $87.0 \pm 46.9$ | $133 \pm 53.6$ | $37.6 \pm 12.8$ | $2.68 \times 10^{8} \pm 1.08 \times 10^{8}$ | $7.98 \times 10^{7} \pm 5.21 \times 10^{7}$ |
| *Alexandrium monilatum (57)* | 47.0–56.8 | 138–163 | 191–225 | 63.0–73.5 | $8.88 \times 10^{8}$–$1.34e \times 10^{9}$ | $4.36 \times 10^{8}$–$6.54 \times 10^{8}$ |
| | $52.0 \pm 19.0$ | $148 \pm 57.7$ | $208 \pm 66.5$ | $68.2 \pm 19.3$ | $1.12 \times 10^{9} \pm 8.66 \times 10^{8}$ | $5.51 \times 10^{8} \pm 4.39 \times 10^{8}$ |
| *Mnemiopsis leidyi (52)* | 190–241 | 935–$1.15 \times 10^{3}$ | $1.13 \times 10^{3}$–$1.35 \times 10^{3}$ | 243-281 | $1.36 \times 10^{12}$–$1.84 \times 10^{12}$ | $6.90 \times 10^{11}$–$9.24 \times 10^{11}$ |
| | $215 \pm 96.0$ | $1.04 \times 10^{3} \pm 393$ | $1.24 \times 10^{3} \pm 397$ | $262 \pm 67.6$ | $1.60 \times 10^{12} \pm 8.71 \times 10^{11}$ | $8.10 \times 10^{11} \pm 4.31 \times 10^{11}$ |
| *Oikopleura sp. (9)* | 33.5–54.6 | 55.6–270 | 90.4–333 | 35.7–104 | $9.40 \times 10^{8}$–$1.89 \times 10^{9}$ | $3.43 \times 10^{8}$–$9.30 \times 10^{8}$ |
| | $44.1 \pm 16.9$ | $162 \pm 173$ | $212 \pm 195$ | $70.4 \pm 56.3$ | $1.41 \times 10^{9} \pm 7.72 \times 10^{8}$ | $6.30 \times 10^{8} \pm 4.89 \times 10^{8}$ |

Figure 3 shows a typical example of a first flash for the seven tested species. Individual emissions were normalized to their respective peak intensities, allowing for better visualization and comparisons of their timing parameters. With the exception of *M. leidyi*, all species reach their PI within 100 ms. *M. leidyi* sustains near PI levels for a longer duration than other tested species, which translates into high FF-MSL (Table 1). Dinoflagellate species have abrupt decay times and e-folding times, however both *Pyrocystis* species tested here have a long exponential decay phase (*Widder & Case, 1981*). The "plateau" sections of *L. polyedra*'s flash response are a consequence of its peak intensity being very close to the noise floor of the UBAT. For this species, the signal to noise ratio is thus much lower than other species tested, resulting in low quantization and a broken decaying phase.

Since one of the goals of this study was to identify which FFKPs can be used to differentiate species from their bioluminescent emissions, non-parametric one-way ANOVAs were applied (Table 2), showing the flash parameter distributions that can be statistically distinguished for every comparisons of tested species. For example, *P. fusiformis* and *P. noctiluca*, peak intensity (PI) and FF-MSL can be used to identify these species when looking at their bioluminescent emissions (Table 2, top-left cell). At least one FFKP could be used for all species comparisons except when comparing *Oikopleura sp.* to *P. bahamense* or *A. monilatum*. All FFKPs of *M. leidyi* could be used to distinguish between bioluminescent emissions of all other test species. When comparing FFKPs of *P. bahamense* and *A. monilatum*, e-folding time was the only statistically relevant parameter. From this species assemblage, peak intensity and FF-MSL were the most consistent flash kinetics to use in order to discriminate specific species. Flash kinetics linked to the timing of a flash

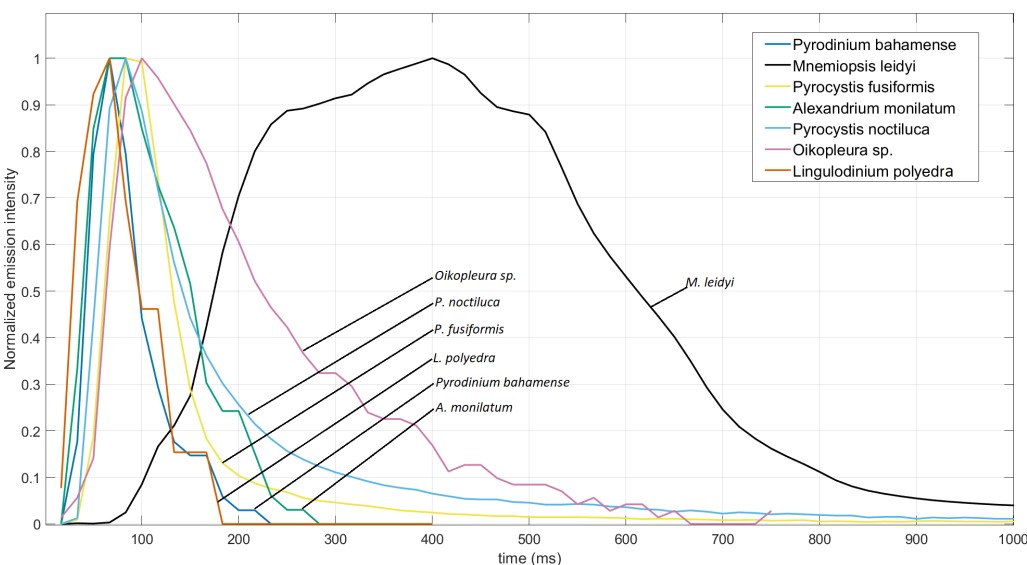

**Figure 3** **Typical first flash response of all tested species normalized to their respective peak intensities.**

**Table 2** **Summary of the non-parametric one-way ANOVA analysis using the Kruskall-Wallis test applied to the flash kinetics of all tested species.** The presence of the flash kinetic's acronym indicates a statistically significant difference in the parameter's distribution at a $p < 0.05$ significance level between the intersecting species. PI-Peak Intensity, FF-First flash mechanically stimulated light, RT-Rise time, DT-Decay Time, FD-Flash Duration, EF-e folding Time.

| Species \ Species | P. fusiformis | P. noctiluca | Pyrodinium bahamense | L. polyedra | A. monilatum | M. leidyi | Oikopleura sp. |
|---|---|---|---|---|---|---|---|
| P. fusiformis | | PI  FF | PI  FF  RT / DT  FD  EF | PI  FF  RT / DT  FD  EF | PI  FF  RT / DT  FD  EF | PI  FF  RT / DT  FD  EF | PI  FF  RT / DT  FD  EF |
| P. noctiluca | | | PI  FF  RT / DT  FD  EF | PI  FF  RT / DT  FD  EF | PI  FF  RT / DT  FD  EF | PI  FF  RT / DT  FD  EF | FF / DT  FD  EF |
| Pyrodinium bahamense | | | | PI  FF | | PI  FF  RT / DT  FD  EF | EF |
| L. polyedra | | | | | PI  FF / DT  FD  EF | PI  FF  RT / DT  FD  EF | PI / DT  FD |
| A monilatum | | | | | | PI  FF  RT / DT  FD  EF | |
| M. leidyi | | | | | | | PI  FF  RT / DT  FD  EF |
| Oikopleura sp. | | | | | | | |

can mostly be used, *i.e.,* FD, RT, DT and EF, but their statistical significance is less reliable, specifically for the dinoflagellate *P. bahamense.*

A hierarchical clustering analysis was performed to assess how FFKP-based clustering would compare to the phylogeny of the tested species. The goal of this analysis was to test if bioluminescent emissions vary in parallel with evolved similarities. Figure 4A shows the cluster analysis and Fig. 4B shows the evolutionary relations of the species. The hierarchical clustering analysis highlights similarities in FFKPs, grouping species

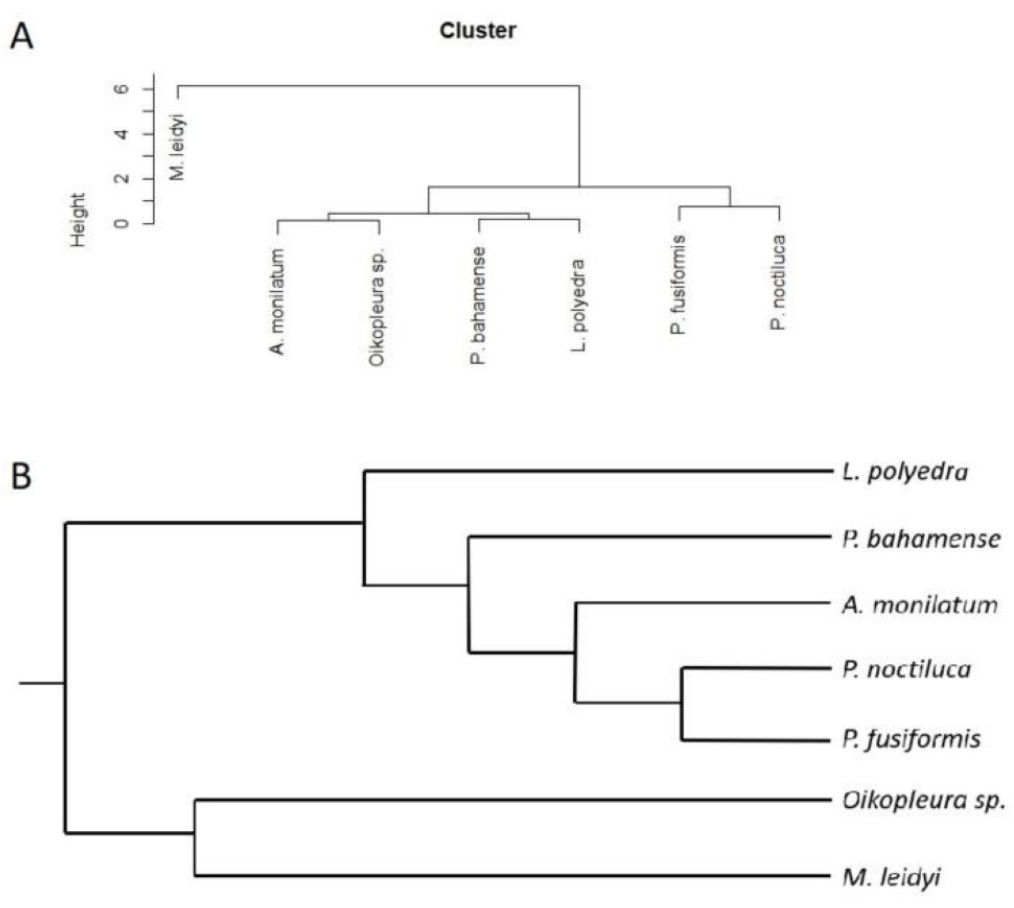

**Figure 4** **(A) Hierarchical clustering analysis of all tested species using flash kinetics as clustering criterion. (B) Phylogenetic tree of tested species, showing their taxonomic relations.** Dinoflagellate phylogeny is based on *Murray et al. (2005)*, phyloT (https://phylot.biobyte.de/) and Interactive Tree of Life (*Letunic & Bork, 2021*).

with similar bioluminescent emission characteristics together, as a phylogenetic tree taxonomilcally groups related species. In both trees, the outlier is the ctenophore *M. leidyi*. The two *Pyrocystis* species are also nested together in the cluster analysis, matching the phylogeny. The largest discrepancy between trees is the nesting of *Oikopleura sp.* within dinoflagellate species in the clustering analysis, since it belongs to the phylum Chordata. However, this warrants a much broader analysis including several additional species from other phyla to fully assess how FFKPs vary in parallel to phylogeny.

Finally, a linear discriminant analysis (LDA) (*Tharwat et al., 2017*) was applied to calculate relative probability of species identification when only the first flash kinetic parameters are known. When the UBAT is deployed in a water column, matching of FFKPs to a specific species requires a library of FFKPs for all sampled species. This exercise is intended as a simulation with only the organisms sampled here. If sampled species are not represented in the FFKP library, predictions will be biased. The LDA model was trained using measured PI, FD, RT, and EF. Figure 5 shows all samples in multivariate space,

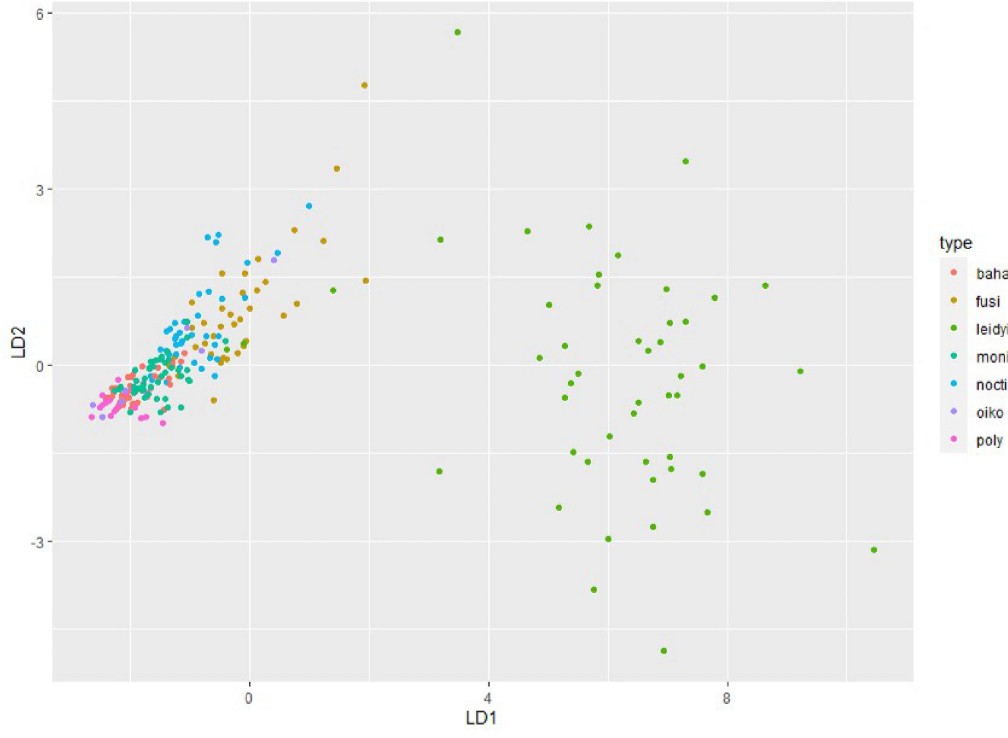

**Figure 5   Visual representation of the linear discriminant analysis of the multivariate space involving PI, FD, RT and EF.** For maximal model accuracy, all species are included except *P. bahamense*. Species acronyms are the same as in Fig. 2.

color-coded by species. LD1 and LD2 had a cumulative explained variance of 99.36%. It is evident the species with most variation is the ctenophore *M. leidyi*. Tested dinoflagellate species are grouped closely together, although it has been previously established they can mostly be distinguished based on their FFKPs (Table 2).

To test the accuracy of species prediction, a subsample of the UBAT experimental dataset independent from the training dataset was tested on the LDA model. Results of this test can be seen in Table 3. Correct predictions of the model can be interpreted as the diagonal line of the matrix, showing the number of times that prediction was made. Rows in this matrix correspond to the true identifications and columns correspond to the prediction made by the LDA. For example, *P. fusiformis* was correctly identified as such 19 times (top-left corner), but four *P. fusiformis* samples were misidentified as *M. leidyi*. Most misidentifications were made mistaking *P. fusiformis* for other species and mistaking *P. noctiluca* for *P. fusiformis*, which are closely related species of the same genus. In this subsample, none of the larvacean samples were correctly identified. All bioluminescent emissions of *M. leidyi* were correctly identified as such. The overall prediction accuracy was 73%. Although this approach needs to be tested in a real ecosystem, this exercise demonstrates UBAT measurements of water containing unknown organisms for which FFKPs have been described could be a tractable method for biodiversity studies.

**Table 3  Validation matrix of the LDA model testing a subsample against the training sample.** The vertical axis shows the model's decision of the highest species probability, whereas the horizontal axis shows the correct species identification. Correct predictions are indicated by red outlines.

|        | fusi | leidyi | moni | nocti | oiko | poly |
|--------|------|--------|------|-------|------|------|
| fusi   | 19   | 4      | 2    | 3     | 1    | 0    |
| leidyi | 0    | 27     | 0    | 0     | 0    | 0    |
| moni   | 1    | 0      | 32   | 4     | 1    | 7    |
| nocti  | 9    | 1      | 3    | 29    | 1    | 0    |
| oiko   | 0    | 0      | 0    | 0     | 0    | 0    |
| poly   | 0    | 0      | 3    | 0     | 4    | 11   |

**Table 4  Example of an output from the LDA model where relative probabilities are calculated for each species from the training sample.** Numbers in the first column indicate the unknown sample ID.

|   | fusi      | leidyi       | moni       | nocti      | oiko       | poly        |
|---|-----------|--------------|------------|------------|------------|-------------|
| 1 | 0.6374306 | 5.923431e-10 | 0.003732485| 0.34932529 | 0.009065506| 0.0004461454|
| 2 | 0.8828046 | 3.681189e-08 | 0.024316393| 0.03371116 | 0.056382790| 0.0027850273|
| 3 | 0.8761987 | 7.406750e-07 | 0.032757612| 0.03848110 | 0.050000898| 0.0025609827|
| 4 | 0.4513617 | 1.212482e-10 | 0.203954722| 0.21202236 | 0.078028555| 0.0546326710|
| 5 | 0.5415710 | 5.379417e-10 | 0.162934997| 0.18648633 | 0.074957660| 0.0340500440|
| 6 | 0.4513683 | 3.245622e-10 | 0.027744932| 0.50013502 | 0.016995865| 0.0037559182|

With this LDA model, it is possible to test unknown sets of flash kinetics and get probabilistic information on its species ID. This model will provide a relative probability of ID for all the species it contains, using relative distance to species clusters in FKKPs multivariate space. For example, in Table 4, unknown sample 2 has an 88.3% probability of being *P. fusiformis* based on its bioluminescent signature acquired in the UBAT. For sample 6 however, the ID is split between a 45 and 50% probability for *P. fusiformis* and *P. noctiluca*, which also means it has a 95% probability of belonging to the *Pyrocystis* genus.

## DISCUSSION

Most dinoflagellate species with described emissions do not have complete sets of FFKPs. *Latz, Nauen & Rohr (2004)* and *Widder & Case (1981)* described the PI ($4.79 \times 10^{10}$ photons/s), RT (10 ms) and DT (200 ms) of *P. fusiformis*. PI measured in this study was $6.64 \times 10^{10} \pm 4.48 \times 10^{10}$ photons/s on average, and rise time and flash duration were $76.8 \pm 31.0$ ms and $536 \pm 288$ ms, respectively. While PI is within the same order of magnitude, RT is much longer than what was previously measured. However, *Latz, Nauen & Rohr (2004)* mechanically stimulated cells in a pipe flow apparatus and *Widder & Case (1981)* used a pulsed solenoid, both of which most likely provide very different levels of shear stress than the UBAT. Multiple flash forms have also been observed with this species, which could explain variation from our results in time kinetics (*Widder & Case, 1981*). The TMSL of *P. bahamense* has been measured at $3.35 \times 10^{8}$ photons/ind (*Biggley et al., 1969*). TMSL involves constant stimulation until exhaustion of bioluminescent emissions and is not a measurement of a single flash response. Thus, this measurement is not comparable

to FF-MSL measured in this study. FFKPs of *L. polyedra* have been measured in several studies (*Biggley et al., 1969*; *Latz, Nauen & Rohr, 2004*), with a complete set of kinetics in *Latz & Lee (1995)*. PI, RT and FD all fall in 95% confidence interval measured in this present effort. Their e-fold time was measured at $56 \pm 6$ ms while our measurements were $37.6 \pm 12.8$ ms, thus being lower while having more variation. The same can be said for DT; *Latz & Lee (1995)* measured $114 \pm 11$ ms and our average was $87.0 \pm 46.9$ ms.

In this study, FFKPs of only individuals of the larvacean *Oikopleura sp.* were described, although it has been previously observed that secreted gelatinous houses can exhibit significantly different flash kinetics (*Galt & Sykes, 1983*; *Galt, Grober & Sykes, 1985*). The authors were also unable to identify to species level, but comparison within the genus is still possible. RT measured in *Galt & Sykes (1983)* ranged from 10 to 24 ms, while our results were significantly longer ($44.1 \pm 16.9$ ms on average). However, our sample size was very small considering the difficulty in finding organisms at sample sites and in maintaining them alive for several hours in lab conditions. FD results were consistent with previous literature for the *Oikopleura* genus (Table 1; *Galt, Grober & Sykes, 1985*). Although larvaceans have been identified as important contributors to water column bioluminescence (*Cronin et al., 2016*; *Martini & Haddock, 2017*) and several species have been identified as bioluminescent (*Poupin, Cussatlegras & Geistdoerfer, 1999*), very little is known on their mechanically stimulated FFKPs, requiring further research.

Lack of information on other bathyphotometers used for measuring *M. leidyi* bioluminescence do not allow for adequate comparisons with present results. Moreover, different bathyphotometers do not agree due to differing levels of mechanical shear and calibration methods (*Letendre et al., 2024*). *Tokarev & Mashukova (2016)* measured its FFKPs with the Svet device, using constant mechanical stimulation to exhaustion. However, this device measures bioluminescent emissions in quanta[1]$cm^{-2}s^{-1}$, and not knowing key parameters like organism to PMT distance obfuscate conversions. This Svet is more akin to an integrating sphere, which most likely produces very different shear stress profiles than the flow-through design of the UBAT (*Mashukova et al., 2023*). The larval stage of *M. leidyi* was recently measured in a UBAT (*Blackburn et al., 2023*). With the cydippid stage, four different flash responses were observed in the UBAT, whereas adults consistently showed a single flash pattern (Fig. 3). Peak intensities of adults are on average 2 orders of magnitude higher (Table 1; *Blackburn et al., 2023*). All time-dependent FFKPs were much longer in adults. For example, FDs were $1.24 \times 10^3 \pm 397$ ms for adults and $471 \pm 98$ ms for cydippids.

Identifying bioluminescent species using their emissions requires careful consideration of methodology. Historically, a wide variety of instruments have been used, each having different residence times, inherent shear stress profiles and flow rates. All these parameters can affect FFKPs (*Latz & Rohr, 2013*). Thus, referencing the same instrument is key when comparing acquired flash kinetics with existing literature. For example, when the dinoflagellate *L. polyedra* was mechanically stimulated in the HIDEX bathyphotometer (*Widder et al., 1993*), the MSL accounted for 94% of the total mechanically stimulated light (TMSL) measured with continuous stimuli in an integrating sphere, whereas only 17% was stimulated in the UBAT (*Latz & Rohr, 2013*). This decrease in efficiency with
the UBAT is most likely due to lower flow rates and shear stress levels experienced in the chamber, especially considering *L. polyedra* has a relatively high shear stress threshold for dinoflagellate species, *i.e.,* 0.3 Pa (*Latz, Nauen & Rohr, 2004*). The low flow rate of the UBAT, *i.e.,* $0.330 \pm 0.05 \mathrm{l\ s^{-1}}$, may cause significant avoidance of larger zooplankton when deployed in the water column. Not only could organisms avoid entrainment, but previous research has shown zooplankton actively avoid profiling instrumentation (*Benoit-Bird et al., 2010*; *Geoffroy et al., 2021*). Bathyphotometers with much higher flow rates can decrease this avoidance issue and provide a more complete assessment of the planktonic community. Results here are thus strictly only comparable to other UBAT sensors used in the same manner, including aspects such as organism preparation.

In our study, when discriminating between two species, PI and FF-MSL were the most reliable FFKPs. Indeed, PI and FF-MSL were statistically different for 18 and 17 out of the 21 possible species combinations, respectively (Table 2). However, these two kinetics were not valid metrics for ID when *P. bahamense* and *A. monilatum* were both present in the simulation, since their distributions were statistically inseperable. In fact, this was the case for all their FFKPs except for e-folding time. Difficulty in telling these species apart could be explained by their evolutionary proximity; the genus *Pyrodinium* is the sister clade of *Alexandrium* (*Murray et al., 2005*). Based on ribosomal RNA sequencing, *Leaw et al. (2005)* suggested the *Pyrodinium* genus was actually nested within *Alexandrium*, thus making it potentially paraphyletic. When measuring bioluminescence *in situ*, it is likely emissions from two or more species are too similar to discriminate, especially if closely related species are present in the same location. Measuring additionnal FFKPs and the spectral properties of emissions could help further discriminate statistically identical species. Another interesting results from the non-parametric ANOVAs was the absence of any relevant FFKPs for ID when comparing the larvacean *Oikopleura sp.* to the dinoflagellates *A. monilatum* and *P. bahamense*. Considering that larvaceans and dinoflagellates utilize a different family of luciferin for the chemical reaction resulting in bioluminescence, *i.e.,* coelentarazine and dinoflagellate luciferin respectively (*Haddock, Moline & Case, 2010*), their FFKP similarity is worth noting and warrants further research. It is possible similar ecological pressure for both clades, *e.g.,* predators, habitat, resulted in converging evolution of bioluminescent emission characteristics. Introducing spectral properties like peak emitted wavelength and bandwidth would likely help differentiating the emissions of these species (*Herring, 1983*; *Widder, Latz & Case, 1983*).

The highest variability in FFKPs was found in the ctenophore *Mnemiopsis leidyi* (Fig. 2). Whereas results for all species of dinoflagellates were similar, although with a certain degree of separation, ctenophores were sparsely distributed in multivariate space (Fig. 5). This high variability could be attributed to many factors. First, these ctenophores were caught in the HBOI channel and not cultured in lab. Even if we consider the 3 h dark adaptation period, there is no way of controlling for their prior light history, nutrition, general physiological state and health, age, etc. All these factors have been found to have an impact on bioluminescence and FFKPs for this species (*Mashukova & Tokarev, 2013*; *Mashukova & Tokarev, 2016*; *Nikolaevich & Vladimirovna, 2016*; *Tokarev, Mashukova & Sibirtsova, 2012*). High variability in FFKPs can also be attributed to a larger body size

range than with dinoflagellates. Since larger body size correlates to a higher amount of available coelentarazine and luciferase, more total photons and instantaneous intensities can be produced upon mechanical stimulation (*Tokarev, Mashukova & Sibirtsova, 2012*). A broader body size distribution can introduce variation in bioluminescent emissions. Additionally, the larger size of ctenophores implies varied levels of shear stress across the organism as it enters the mixing chamber of the UBAT. Then, photocytes are mechanically stimulated at different intensities, which also contributes to variation in light produced.

## UBAT calibration

To prevent biases in photons collection by PMTs caused by varying distance from the emitter, the UBAT measures bioluminescent flashes in an integrating chamber. These chambers are coated with highly reflective materials, which reflects light in all directions and transforms the light organs into almost perfectly isotropic sources. In theory, this means that the light being emitted is evenly distributed and projected unto the $4\pi$ surface of the integrating cavity. However, when calibrating the UBAT using our own calibrated constant emission standard, we observed up to 30% variation of PMT readings when changing the distance and orientation of the calibrated source (Fig. 6). Thus, variance in bioluminescence measurements is introduced here since organisms are not made into true isotropic sources by the integrating cavity and will not be stimulated at the same location due to the turbulent flow of the UBAT. PI and FF-MSL are the two FFKPs that are most affected by this finding, as radiant flux collected by the PMT will diminish as distance increases. The timing parameters of the bioluminescent flash, however, are not affected by this variation.

Following an independent calibration of the UBAT using a calibrated tritium emitter, it was discovered that the manufacturers' calibration was off by a factor of 2.5 (*Blackburn et al., 2023*). At this time, it is unclear if other UBATs have the same calibration offset, or if this 2.5 factor is variable across instruments. This finding can potentially have numerous ramifications, as UBATs have been used extensively in bioluminescence research (*Cronin, 2015*; *Johnsen et al., 2014*) and flash kinetics measurements of individual organisms (*Krohn-Pettersen, 2023*). This could also mean recent studies have significantly underestimated bathyphotometer-mechanically stimulated light (BP-MSL). Independent calibrations of other UBATs should be made to assess this calibration issue.

Another possible source of variability in ctenophore FFKPs is the prestimulation of the individual in the UBAT's light baffle. Indeed, the UBAT has a light baffle to prevent the introduction of environmental light into the integration chamber and ends with the impeller responsible for creating the mechanical stimulation. During testing, bioluminescence was observed while the individuals were entering the light baffle for most individuals. With *M. leidyi* being significantly larger than dinoflagellate cells, the periphery of the ctenophore experiences high shear at the very entrance of the input, when the individual is entrained and deformed by the flow. This pre-stimulation emission is certainly lost and not collected by the PMT, also introducing variation in measurements.

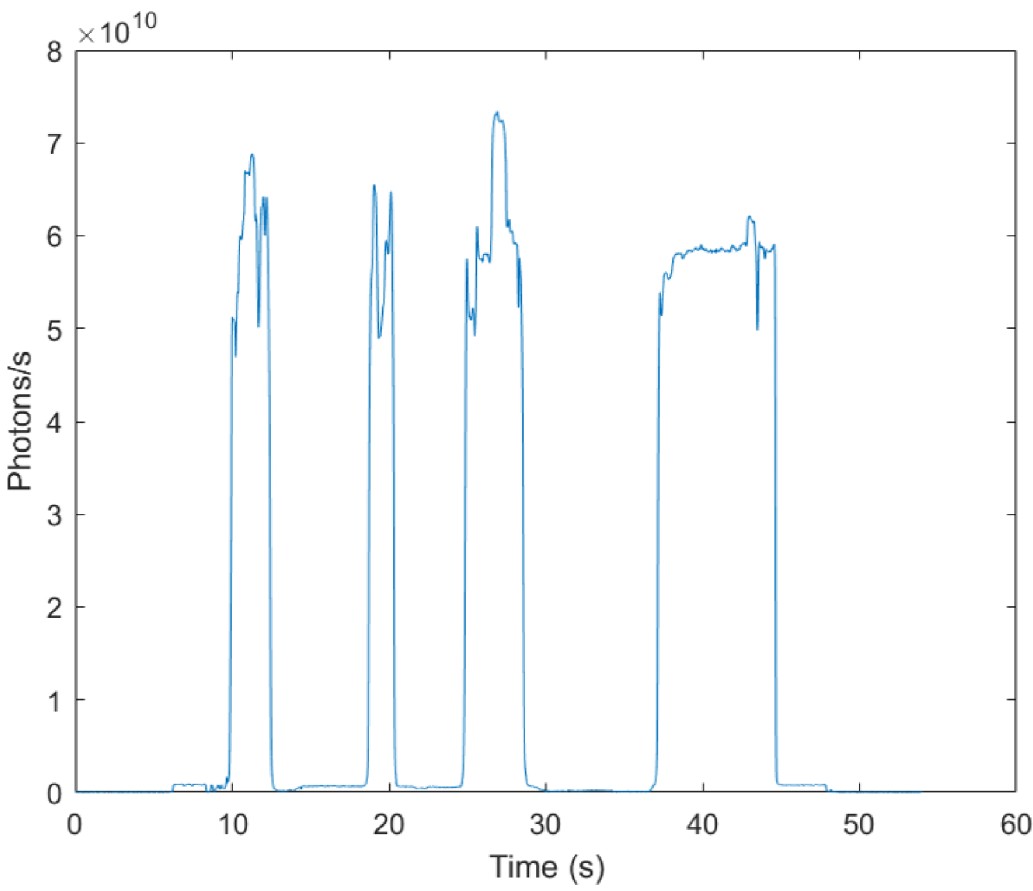

**Figure 6** Variation of the light collected by the PMT inside the UBAT integrating cavity caused by moving the calibrated constant emitter within the chamber. The calibrated radiant flux of the constant emitter was measured at $1.45 \times 10^{11}$ photons/s in an integrating sphere system. Intensity drops to zero are caused by manually turning off the UBAT PMT to repeat measurements.

## Ecological and biological considerations on FFKPs

While this study aims to provide precise FFKPs acquired through single cell isolation and laboratory measurements, bioluminescent signatures measured *in situ* will incorporate prior life and light history biases. In an effort to provide a library of FFKPs that can be used with field-acquired data, bootstrapped confidence intervals were calculated. These intervals will ideally encompass some variations caused by environmental and biotic variables in the water column. When measuring bioluminescence *in situ*, the FFKPs most likely to vary are PI and FF-MSL. Indeed, if the organism is not allowed to rest between mechanical stimulation events, stocks of luciferin will not be able to replenish completely and will result in lower peak intensities and amounts of photons produced (*Latz, Bowlby & Case, 1990*; *Widder & Case, 1981*). *Latz, Bowlby & Case (1990)* observed a reduction to 14–38% of TMSL when three species of copepods were tested 1 h following specimen collection. This decrease in available photons is attributed to stimulation during handling and collection, causing lower peak intensities and fewer flashes. Since it is impossible to

control for prior bioluminescent activity from organisms in nature, one cannot assume mechanical stimulation did not occur recently before the *in situ* measurement.

The life stage of the organism must also be taken into consideration, as FF-MSL has been observed to vary orders of magnitude throughout ontogeny of the copepod *Metridia lucens* and the ctenophore *M. leidyi* (*Batchelder & Swift, 1989*; *Tokarev, Mashukova & Sibirtsova, 2012*). Photoinhibition can also alter bioluminescent emissions. *Sullivan & Swift (1994)* observed a 90% decrease in FF-MSL when the dinoflagellate *Tripos fusus* received blue light irradiance during scotophase. Organisms collected from the water column must be allowed to rest and dark-adapted for 24 h before any measurements of FFKPs. Regionality and environmental parameters such as water temperature and nutrients levels may also affect FFKPs (*Letendre et al., 2024*). For example, the North American population of the copepod *Oncaea conifera* has longer RT and FD than then Mediterranean population (*Herring et al., 1993*). The PI of Black Sea *M. leidyi* peaks at water temperatures of 26 °C (*Olga & Yuriy, 2012*). Additionally, levels of water column hypoxia and the organism's regeneration state impact its FFKPs (*Mashukova et al., 2023*; *Nikolaevich & Vladimirovna, 2016*).

Confidence intervals of FFKPs (Table 1) could help resolve *in situ* identifications of species exhibiting multiple first flash responses. For example, *Widder & Case (1981)* described two flash responses for *P. fusiformis*. Initial stimulation produces a flash with a short rise and decay time, with any subsequent stimulation having a lower peak intensity, and a decay time up to 2.5 times longer than the initial flash. The copepod *Pleuromamma xiphias* exhibits two flash responses upon a single stimulation, a fast emission *via* internal bioluminescence in light organs, and a slow emission *via* released clouds of bioluminescent material (*Latz et al., 1987*). These two emissions can be seen as one in a bathyphotometer if produced simultaneously, or as two distinct flashes with the exuded clouds lagging a few milliseconds behind (*Latz, Bowlby & Case, 1990*). Species able to emit multiple first flash signatures may be more difficult to identify *in situ*, and will most likely require simultaneous measurement of spectral properties.

Adding spectral properties of the tested species into the LDA model would most certainly increase its strength and allow for further differentiation and higher ID accuracy. While most dinoflagellate species with described spectral properties emit in the 470–480 nm range (*Herring, 1983*; *Latz, Nauen & Rohr, 2004*; *Poupin, Cussatlegras & Geistdoerfer, 1999*), bioluminescent emissions ranging from 435 to 583 nm have been measured across planktonic species of diverse phyla (*Herring, 1983*; *Haddock et al., 2005*). However, the bandwidth at half maxima is constrained to 50–100 nm for most species (*Letendre et al., 2024*; *Widder, 2010*). Additionally, spectral properties are independent of factors like light history, diet, etc. Since these are inherent properties of the luciferin protein, spectral properties are most likely constant at the species level, unlike FFKPs.

In this present analysis, the samples of *P. bahamense* were removed from the LDA model since it significantly decreased its ability to accurately identify species. This is most likely due to *P. bahamense* and *A. monilatum* having very similar FFKPs, *i.e.,* only e-fold time was significantly different in non-parametric ANOVAs (Table 2). DT and FF-MSL were also not included in the LDA model since they degraded accuracy. This can be explained by the high level of correlation these variables have with other measured FFKPs. PI and FF-MSL

are strongly related ($r = 0.998$) since bioluminescent flashes are generally very short in duration and most of the photons measured in the FF-MSL metric are from the peak itself. Similarly, DT and FD have a very high correlation ($r = 0.954$), thus being redundant in the linear discriminant analysis. While the LDA model had a 73% accuracy, some of these misidentifications were made by confusing two species of the same genus (Fig. 4). With this in mind, this model could have higher accuracy if only the genus information is needed (82%).

With a robust and fully described library for key periods of a given ecosystem, this type of analysis can provide biodiversity information *in situ* while limiting the need for laborious traditional sampling methods. Once a planktonic community has had its bioluminescent signatures described in controlled laboratory settings, an LDA model like the one developed in this study could be applied to ID bioluminescent organisms autonomously and assess a system's community composition over a wide range of spatial and temporal scales. Moreover, in most regions, few species dominate water column bioluminescence, making this type of analysis a practical and tractable one. This type of analysis with sensors deployed from autonomous underwater vehicles (*Berge et al., 2012*; *Moline et al., 2005*), profilers (*Moline et al., 2002*) and buoys (*Lapota et al., 2002*; *Lapota, 2005*) could prove to be a powerful tool for monitoring marine diversity and biomass once a comprehensive emission kinetics library has been compiled. Mixed signals from multiple organisms and species flashing in flow-through bathyphotometers can be separated and identified using an empirical orthogonal function analysis (*Davis et al., 2005*). Bioluminescent signatures have been used to describe planktonic communities from the Arctic (*Cronin et al., 2016*; *Johnsen et al., 2014*) and from Monterey Bay, California (*Moline et al., 2005*) using similar methods, where known species had their FFKPs described in the lab using the UBAT. This library was then referenced when testing a natural sample of unknown organisms captured in net tows. An autonomous bathyphotometer referencing an established library of FFKPs would not only provide presence/absence data, but may also provide diversity indices on planktonic species partitioning within the water column.

## ACKNOWLEDGEMENTS

The authors sincerely thank Michael I. Latz for his guidance on statistical analyses, and Trevor McKenzie and Juan Aguilar for their help on data visualization. The authors are also grateful for the insightful comments of the three reviewers and editor, which significantly improved the quality of the manuscript.

### Funding

Funding was provided by a Florida Atlantic University postdoc fellowship grant. The funders had no role in study design, data collection and analysis, decision to publish, or preparation of the manuscript.

## Grant Disclosures

The following grant information was disclosed by the authors:
Florida Atlantic University.

## Competing Interests

The authors declare there are no competing interests.

## Author Contributions

- Francis Letendre conceived and designed the experiments, performed the experiments, analyzed the data, prepared figures and/or tables, authored or reviewed drafts of the article, and approved the final draft.
- Abigail Blackburn conceived and designed the experiments, performed the experiments, prepared figures and/or tables, and approved the final draft.
- Ed Malkiel conceived and designed the experiments, prepared figures and/or tables, and approved the final draft.
- Michael Twardowski conceived and designed the experiments, authored or reviewed drafts of the article, and approved the final draft.

## Data Availability

   All kinetics measured in the UBAT, raw data and processing codes are available in the Supplementary Files.

## Supplemental Information

Supplemental information for this article can be found online at http://dx.doi.org/10.7717/peerj.17516#supplemental-information.

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
