# Peer review of "Using bioluminescence as a tool for studying diversity in marine zooplankton and dinoflagellates: an initial assessment"

_PeerJ, doi:10.7717/peerj.17516_

## Round 0.1 · original submission · Major Revisions

All three reviewers (and myself) agree that this is an interesting study deserving publication. However, they have a number of comments that need to be addressed before the paper can be accepted. See their comments below. Here I add a few more comments.

Now that your recent Front. Mar. Sci. paper is published, it would be useful to cite it to put these 2 studies in context of each other. Fig. 1 is also very similar to Fig. 1 in that paper, but different (peak intensity is notably different) despite both being labeled as "Typical first flash kinetic parameters of the dinoflagellate Pyrocystis fusiformis". It could be more informative to display a different species instead, and add in the figure legend a sentence referring the reader to the Front. Mar. Sci. Fig. 1 for a similar example using Pyrocystis fusiformis.

The discussion could also benefit from clearer structuring (i.e., in subsections) particularly if expanded following reviewers' suggestions.

Last, out of curiosity, why is the L. polyedra flash line in Fig. 3 so "broken"? (sharp transitions and constant emission for 10s of ms). Is this truly a typical example for this species?

Please provide a detailed point-by-point reply to the reviewers' comments, ensuring that they are all properly addressed.

I look forward to receiving the revised manuscript.

Reviewer 1 ·

Basic reporting

'no comment'

Experimental design

The paper « Using bioluminescence as a tool for studying diversity in marine zooplankton and dinoflagellates” proposes an analysis of several dinoflagellate and zooplankton species bioluminescence kinetic emission in order to describe diversity of bioluminescent organisms without traditional net methods.
At first, the idea is innovative and objectives are of great importance since the authors propose new methods to detect and characterize planktonic communities in an automated way.
However, in the current way, I have issues to be convinced in some aspect of the method and its future applications.
My main concern is about the application of the method on experimental recorded peaks. Are FFKPs parameters estimated by hand? how do you deal with the sampling rate not fully catching the peaks with high definition? (for example in Fig. 3 what is included into the Lingulodinum polyedra FFMSL parameter – until 190ms? why is the beginning not starting at lower values? )
Abstract :
L16 : not all, « most of the bioluminescence organisms”
L25-27: “it is possible to detect presence without traditional plankton sampling methods […] may be possible”. This sentence seems optimistic, I am not convinced traditional sampling method can be avoided as preliminary knowledge in the area observed with UBAT. How will it be possible to relate peaks of bioluminescence with biomass? This is not clear to me after reading this paper.
Introduction
In my opinion, the introduction could be better organized and it would help to clarify what concerns the surface, the water column, phytoplankton or zooplankton.
L95: “since … species-specific”. While there is some literature supporting classification of bioluminescent organisms based on their peaks I think this first sentence is a very strong hypothesis in front of the wide diversity of luminous organisms and only few being tested in the cited literature.
Results:
L184: “in general” not clear to me
L191: how many of them could not be measured (total vs luminous ones)?
Figure 2: Why not all parameters are presented here?
Please modify units (msec to ms)
Table 1: Why are there some differences in sample size between Fig2 and Table 1?
Fig 3: What does “typical” example means? how were they selected? Why not to use the average instead?
This figure really question myself on how the parameters are defined based on these peaks. Maybe a few peaks with parameters measurements on each could help understand the effectiveness of the method? or eventually use a model of these peak to have continuous values? I am concerned FF-MSL, beginning and end of each peak seem challenging and could have an impact on parameters.
Fig 4: The taxonomic analysis is based on few species only. I am not an expert but I am wondering how the low number of species and diversity affect such analysis.
L237 and 250: Table instead of Figure
Table 3 and 4: what is the impact of a luminous organism not in the library? It will be identified as something existing most likely but without plankton net there is no way to detect it I guess ?

Discussion:
L351: are tritium and calibration LED of same wavelength? Is there an impact depending on PMT wavelength sensitivity spectra?
L399: “435 to 583nm spectra” is true but very narrow for most species (widder 2010)

I would love to see a few more important points in the discussion:
- The authors don’t discuss how the environment could disturb the signal (directly or indirectly link with organisms’ physiology).
- In area with high concentration of organisms several cells could be pumped into the UBAT and the peaks integrated, I guess?
L415: seasonality involves changes in species composition, or abundance. Is the method robust to changes in species composition? Without plankton nets, we would potentially miss some changes if the method is only based on an already acquired library.

Validity of the findings

Conclusions seem wide compared to the study results (biomass estimation).

Underlying dataset on how the parameters are applied on the measured peaks are missing to evaluate their robustness.

Reviewer 2 ·

Basic reporting

no comment

Experimental design

no comment

Validity of the findings

no comment

Additional comments

This manuscript made an in-depth analysis on flash kinetics of several species of dinoflagellates and two zooplankton, and established their first flash kinetic parameters (FFKP). It is very important for discriminating plankton and studying the diversity and ecological system by using bioluminescence in future. It deserves publication. But some questions must be resolved. First, how to ensure the growth phase are the same for one specific species when measuring? In fact, not only the growth phase would affect the measurements of physiological status of the dinoflagellates, the environmental factors such as temperature will also change the flash. So, I suggest the authors add some complementary analysis on this point. Second, the authors said for the first FFKP could discriminate the species, but for the in-situ measurements, all the signals from different species are merged together, how to establish the FFKP for one species, and how to extract the FFKP from the merged signals and further use the method proposed by the authors to discriminate the species, if these were not well resolved, it is very difficult to use in practice. Besides, ANOVAs should be given the full name when it is first refered in the abstract and text. And the second paragraph and the third paragraph should be merged into one paragraph.

Reviewer 3 ·

Basic reporting

The language or your manuscript is generally clear, professional and pleasant to read. I spotted some minor spelling errors which should be fixed:
- Line 215 spelling error: flask kinetics (same error in the caption of Table 2.)
- Line 237 Fig 3 is referred instead of Table 3
- Line 250 Fig 4 is referred instead of Table 4
- Line 250 FKKs should be FFKPs?

Your figures and tables are relevant and show generally good quality. However, I have some suggestions to improve your presentation. Figure 2 should show data for all first flash kinetics parameters, or you should explain why only time-dependent parameters are shown. Additionally, it should be explained more clearly in Table 4 what the numbers 1-6 (first column) stand for. Table 3 would be easier to read if the correct predictions would be highlighted.

It is good that you provide the extracted FFKP values for all individuals, but also the raw data from UBAT measurements as well as the analyzing codes should be made available.

You mention size as a possible reason for intraspecific FFKP variation in M. leidyi. If you have measured the size of the individuals, your manuscript would benefit from a formal analysis of the effect of size on the FFKP.

Experimental design

The background of your work is well defined, but you could express the research question and knowledge gap to fill more clearly. For example, why exactly these species were chosen for the study, how well do they represent the diversity in the area?

You have described the methods mostly very well, and I was pleased to see that you had taken special care to ensure the accuracy of the methodology related to UBAT calibration. Sample sizes are generally good, and you gave a valid reason for low number of the larvacean Oikopleura sp. specimens. One methodological detail, which is not clearly stated is the flash signal thresholding. You tell in lines 90-92 that signal start and end time were based on instrument baseline, but nothing is said about the noise filtering. This is a critical analyzing step especially when moving to work with in situ data. Your data may be so clean that no thresholding or filtering is needed, but it is impossible to evaluate without seeing the raw data and flash extraction code.

You are describing in lines 145-146 that the UBAT was submerged in a black bin. The ambient light conditions during measurements could be described a bit more detailed, since the UBAT is not totally lightproof. Was there a lid or cover on the bin?

Validity of the findings

The findings presented in this manuscript seem to be solid and you have discussed properly both the limitations and strengths of the dataset giving a good perspective on how the findings fit in the existing literature. However, lack of raw data and analyzing codes makes it difficult to evaluate the robustness of the data.

Your conclusions are commendable, but some statements are maybe a bit too bold. For example lines 245-247: “This exercise demonstrates UBAT measurements of water containing unknown organisms taken in locations where FFKPs have been described can be used to provide relatively high probabilities of species ID.” I think this should be stated a bit more carefully, since your result is based on a very limited mock community and not on an actual water sample. Also, I would consider changing the wording in your title, since the zooplankton part is not very comprehensive.

In the discussion, you have considered well many aspects which could differentiate the results of in situ data from your carefully controlled laboratory experiments, such as light and life history. I would like your discussion to encompass the signal processing difficulties such as ambient light reaching the instrument and multiple organisms entering the device simultaneously as well.

Additional comments

Your manuscript Using bioluminescence as a tool for studying diversity in marine zooplankton and dinoflagellates presents a piece of work with good quality. Your work and especially the LDA model is a step forward in implementing laboratory-derived bioluminescence parameters to in situ identification of species/taxa. However, the method needs quite a lot of development, and you should be carful not to be too bold in your statements.

Here listed a couple of additional comments on the manuscript:
- Line 86: “Bioluminescent plankton emit light mainly through mechanical stimulation.” Please provide a reference.
- Lines 197-199: It is stated that “Interestingly, most of the FF-MSLs are 2 orders of magnitude lower than their corresponding peak intensities for a given species.”. The meaning of this sentence is not quite clear, since these two parameters can’t be compared to each other in a straightforward manner. Please, clarify what this means.
- In you discussion you state that TMSL is not comparable with FFKP:s, which is a fair statement. Could you elaborate more on how the first flash is defined and can M. leidyi results be referred as FFKP:s if it gets triggered at several spots one after each other?

---

## Round 0.2 · Minor Revisions

Thank you for the revised version of your manuscript. It was reviewed by Reviewer #3 who agreed that their comments were properly addressed. Unfortunately, Reviewers #1 and #2 were not available, but I believe their comments were properly addressed. Since they could not comment on this new version of the manuscript, here are a few minor comments.

- abstract: "The bioluminescence of the system is often a good proxy for zooplanktonic biomass" - I don't think this is correct, because bioluminescence can be dominated in some cases by autotrophic dinoflagellates. While these are often mixotrophic, they may not be representative of zooplankton. "Planktonic" would be OK.
- l. 110 "that often found": did you mean "that were often found"?
- l. 158: the function is "findpeaks" not "findpeak" according to your code; also please clarify its origin (eg "the findpeaks.m native Matlab function" or similar, since there are other functions of the same name on Matlab Exchange).
- l.164-168: please revise this sentence for missing/extra words (eg "low too low" and there may be a verb missing - consider breaking into 2 sentences for clarity).
- Fig. 3: Thank you for your explanation of the reason for L. polyedra's "plateau" sections. Please mention this briefly in text (or in the figure caption) as it is a useful point and other readers may wonder about it.
- Fig. 6 caption: please provide the light intensity emitted by the constant emitter. Was it on the whole time, or do times when the intensity drops to zero coincide with times when the emitter was moved to a new location? It isn't clear whether the variability discussed l. 364-368 is between ~ 4.5-7.5E10 ph/sec, or includes the drops below 1E10.
- l. 373: it would be useful to cite Blackburn et al (2023) again to clarify this is not a result of this paper.
- Response to Rev #2: "Tested organisms were picked from the newest transfers/generations, increasing probability of testing individuals of similar sizes and ages within one species." - this is relevant and should be added to the manuscript.

Please address these comments by correcting the manuscript accordingly and providing a detailed point-by-point reply.

As a side note (not asking for any change, just a personal comment) I share Rev 1 & 2 concern that merged signals will be an issue in some regions. While the method by Davis et al. (2005) is interesting, it seems unlikely to work in places with high concentrations of bioluminescent plankton such as coastal areas. See for instance Fig. 2c and 3a in Messié et al. (2019): with concentrations ~ 200 cells/L, the bioluminescence being measured is several times higher than PI and flashes become indistinguishable (and higher concentrations are often observed for dinoflagellates).

I look forward to receiving the revised manuscript.

Reviewer 3 ·

Basic reporting

No comment

Experimental design

No comment

Validity of the findings

No comment

Additional comments

The authors have sufficiently addressed all the issues I brought up at the first round of review.

---

## Round 0.3 · accepted · Accept

Thank you for incorporating the final revisions. I am happy to approve this manuscript for publication.